# Understanding the Multidimensional Effects of Polymorphism, Particle Size and Processing for D-Mannitol Powders

**DOI:** 10.3390/pharmaceutics14102128

**Published:** 2022-10-07

**Authors:** Lena Mareczek, Carolin Riehl, Meike Harms, Stephan Reichl

**Affiliations:** 1Institute of Pharmaceutical Technology and Biopharmaceutics, Technische Universität Braunschweig, 38106 Braunschweig, Germany; 2Department of Pharmaceutical Technologies, Merck KGaA, 64293 Darmstadt, Germany

**Keywords:** mannitol, polymorphism, roller compaction, direct compression, powder characterization, surface area, processability, tabletability

## Abstract

The relevance of the polymorphic form, particle size, and processing of mannitol for the mechanical properties of solid oral dosage forms was examined. Thus, particle and powder properties of spray granulated β D-mannitol, β D-mannitol, and δ D-mannitol were assessed in this study with regards to their manufacturability. D-mannitol is a commonly used excipient in pharmaceutical formulations, especially in oral solid dosage forms, and can be crystallized as three polymorphic forms, of which β is the thermodynamically most stable form and δ is a kinetically stabilized polymorph. A systematic analysis of the powders as starting materials and their respective roller compacted granules is presented to elucidate the multidimensional effects of powder and granules characteristics such as polymorphic form, particle size, and preprocessing on the resulting tablets’ mechanical properties. In direct compression and after roller compaction, δ polymorph displayed superior tableting properties over β mannitol, but was outperformed by spray granulated β mannitol. This could be primarily correlated to the higher specific surface area, leading to higher bonding area and more interparticle bonds within the tablet. Hence, it was shown that surface characteristics and preprocessing can prevail over the impact of polymorphism on manufacturability for oral solid dosage forms.

## 1. Introduction

It is well-known that different polymorphs of the same chemical substance can have a profound impact on pharmaceutical development, and therefore, the crystal form of a solid material is often considered a critical quality attribute [1,2]. The different crystal arrangements of the polymorphs lead to different molecular packing and inter- and intramolecular interactions that often result in diverse physico-chemical properties like chemical stability, solubility, density, hygroscopicity, or melting point [3,4]. These differences can substantially influence not only a formulation’s performance, as reported for the lactose pseudopolymorphs as dry powder inhalation carrier [5], but also a formulation’s manufacturability and mechanical properties, such as tablet tensile strength [6].

The thermodynamically most stable form of the active pharmaceutical ingredient (API) is usually preferred in pharmaceutical development in order to minimize the risk of polymorphic transition and the associated possible differences in material properties as they may affect stability, quality of performance, and manufacturability of the drug product. The International Council for Harmonisation of Technical Requirements for Pharmaceuticals for Human Use (ICH) topic Q6A specifications gives guidance on test procedures and acceptance criteria for new chemical drug substances and products, and addresses if and how acceptance criteria for polymorphism should be defined [7].

As excipients can have a wide range of applications in pharmaceutical development, the critical material attributes of an excipient are highly dependent on the specific product and manufacturing process. Therefore, in addition to the chemical, physical, microbiological, and identity requirements, there are also functionality-related characteristics (FRC) included in the excipient monographs of the European Pharmacopoeia. FRCs provide guidance on the definition of specifications based on standard analysis procedures but they have a non-mandatory character [8]. Examples of such functionality-related excipient characteristics are the particle size distribution, loss on drying and flowability of microcrystalline cellulose [9], or bulk and tapped density for lactose monohydrate [10]. In accordance with these regulations, it is also possible to use several polymorphic forms of one excipient with different properties for formulation development.

Although the polymorphs of many pharmaceutical excipients and their polymorphic transformation have been well described [11,12,13,14], there is still little literature on the influence of excipient polymorphism on manufacturability of oral solid dosage forms. Excipients such as, for example, fillers, often represent the largest share of a formulation, and as such their properties can have a significant impact on the properties of the complete formulation. Therefore, it is of great interest to determine how polymorphism in excipients influences the formulation properties and processing of oral solid dosage forms.

This study deploys mannitol and its polymorphic forms, as it is a commonly used filler and binder for oral solid dosage forms due to its high solubility, low drug-excipient interaction tendency, non-hygroscopic character, and sweet taste [15].

Three polymorphs (α, β, δ) and a monohydrate of mannitol are known with β as the most stable and δ as the least stable polymorph at room temperature. As β is the thermodynamically most stable form, it is also the most frequently used. Despite its thermodynamic metastability, the δ polymorph demonstrates a kinetic stability and therefore, when stored under dry conditions, does not undergo polymorphic transition over years [16]. For various manufacturing processes, improved mechanical compression properties for δ polymorph compared to the thermodynamically stable β polymorph were reported. Yoshinari et al. could show better tableting properties of δ mannitol after wet granulation compared to initial β mannitol, which they attributed to a change in particle morphology due to polymorphic transition [17]. These tableting advantages for δ mannitol compared to β mannitol were also demonstrated after twin screw granulation of binary mixtures of the mannitol polymorphs and acetaminophen by Vanhoorne et al. [18,19]. Roller compaction processability of unprocessed δ and β mannitol, as well as preprocessed β mannitol, was investigated by Wagner et al. [20,21], where it was shown that δ mannitol exhibited beneficial tableting properties after roller compaction compared to unprocessed β mannitol. However, these studies especially focused on the respective processes. However, the underlying material properties of the polymorphs are an important part of understanding processability differences in mannitol and need to be addressed in more detail. The complexity of the many connections between different material properties creates a challenge in determining key material attributes for processing. Therefore, this study aims to gain deeper knowledge about the root causes for the differences in processability of the mannitol polymorphs by systematically investigating the material attributes of the polymorphs like the individual impact of particle size, morphology, and the processing route by an in-depth comparison of direct compression and roller compaction. Furthermore, this study focuses on the mechanical properties of mannitol, like the elastic–plastic behavior, and their connection to the powder properties in order to better explain the differences of the mannitols during tableting.

Direct compression and roller compaction were considered especially relevant for this study and the pharmaceutical industry, because, as stated in the Manufacturing Classification System (MCS) for oral solid dosage forms by Leane et al. [22], these two processing routes have the lowest process complexity, yet the highest requirements for the material properties compared to the other classes. Mechanical characteristics of the mannitol polymorphs can be well-examined in direct compression and roller compaction as the material is exposed to different degrees of mechanical stress.

This study aims to elucidate the influence of the polymorphic form of mannitol on the mechanical properties of oral solid dosage forms by systematically analyzing powder and particle properties of the thermodynamically stable β mannitol polymorph and the kinetically stable δ mannitol polymorph in two different manufacturing processes: direct compression and roller compaction (RC). Various powder characteristics, such as particle size, density, flowability, powder rheology and compressibility, surface area, and particle morphology, were studied to gain deeper insight into the underlying factors affecting the mechanical properties of the polymorphs during tableting. Subsequently, the authors expanded the systematic assessment for two key influencing factors which are essential in the excipient selection to better decipher the multidimensional effects on powder tabletability. First, two different particle size distributions of β mannitol were examined to specifically evaluate the impact of particle size on mechanical properties of the polymorphs. Second, the authors compared unprocessed β and δ mannitol to spray granulated β mannitol to assess the contribution of particle morphology to the powder and mechanical tableting characteristics of the two mannitol polymorphs.

## 2. Materials and Methods

### 2.1. Materials

β mannitol (Pearlitol^®^ 160C) was acquired from Roquette Frères (Lestrem, France), and δ mannitol (Parteck^®^ Delta M) and spray granulated β mannitol (Parteck^®^ M200) were supplied by Merck KGaA (Darmstadt, Germany). Magnesium stearate was purchased from Peter Greven (Bad Münstereifel, Germany). All powders were sieved through a 1 mm sieve using a Turbosieve BTS 100 (L.B. Bohle, Ennigerloh, Germany) with a speed of 355 rpm. The influence of initial powder particle size on granules and tablet properties was investigated by comparison of two different particle size distributions of β mannitol powder: the initial 1 mm turbosieved powder and a fraction smaller than 180 µm of the β mannitol (180 β mannitol) powder. Therefore, after sieving with the turbosieve, a fraction of Pearlitol^®^ 160C powder was again manually sieved through a sieve with a mesh size of 180 µm (Retsch GmbH, Haan, Germany) before further processing.

### 2.2. Methods

#### 2.2.1. Roller Compaction

Granules were obtained by roller compaction on the Mini-Pactor^®^ (Gerteis, Rapperswil-Jona, Switzerland) using smooth rolls with a diameter of 25 cm and width of 2.5 cm. A specific compaction force of 9 kN/cm, gap width of 3 mm, and a roll speed of 3 rpm were set as fixed process parameters to enable systematic comparison. An oscillating star-granulator equipped with a 1.0 mm sieve was utilized for ribbon granulation.

#### 2.2.2. Particle Size Distribution

Particle size distribution of powders and granules was assessed by dynamic image analysis using the X-Jet module of the CamSizer X2^®^ (Retsch GmbH, Haan, Germany) with a dispersing air pressure of 25 kPa to achieve dispersion of the particles without causing breakage of the granules. Measurements were conducted in triplicate, and samples were divided with the sample splitter RT 12.5 (Retsch GmbH, Haan, Germany) before measurement. Cumulative particle size distribution curves are shown in Appendix A.

#### 2.2.3. Powder Densities

Pycnometric density (ρ_p_) of the powders and granules was evaluated with nitrogen via gas displacement technique using an Ultrapyc 1200e (Quantachrome Instruments, Boynton Beach, FL, USA). A test cell with a volume of 58 cm³ was filled to at least 80% with the sample powder, and sample mass was weighed. The arithmetic means of *n* = 3 ± S.D. are given in the Appendix A.

Bulk (ρ_b_) and tapped (ρ_t_) density were determined in triplicate using the Granupack™ (GranuTools, Awans, France), applying 500 taps and a tapping frequency of 1 Hz. A powder sample of known mass (m) was poured into a steel cylinder, and initial volume (V_0_) and volume after tapping (V_t_) were determined. Bulk and tapped density were calculated as m/V_0_ and m/V_t_, respectively.

#### 2.2.4. Flowability

Flowability, expressed as the flow function coefficient (ffc) of the powders and granules, was determined in triplicate by a ring shear tester RST-XS (Dietmar Schulze Schüttgutmesstechnik, Wolfenbüttel, Germany). A normal preshear stress of 9000 Pa was applied and the samples were sheared under three different consolidation stresses: 1800, 4500, and 7200 Pa.

Utilizing straight line regression, ffc was calculated as a ratio of major principal stress (MPS) and unconfined yield strength (UYS), and evaluated according to Jenike [23].
(1)$$\mathrm{ffc}=\frac{\mathrm{major}\;\mathrm{principle}\;\mathrm{stress}}{\mathrm{unconfined}\;\mathrm{yield}\;\mathrm{strength}}$$

#### 2.2.5. Powder Compressibility

Measurement of powder compressibility as *v*/*v*% reduction under pressure was performed in triplicate using the FT4 powder rheometer (Freeman Technology, Tewkesbury, UK) according to established protocols [24,25,26]. Powder compressibility was measured in a glass vessel with a diameter of 50 mm at varying normal forces applied by a vented piston. The relative change (*v*/*v*%) in the powders’ volume based on the uncompacted, conditioned state was determined for each of the applied compression forces in the range of 0.5 kPa to 15 kPa, with each load held being for 60 s.

#### 2.2.6. Specific Surface Area

Specific surface area (SSA) was determined via inverse gas chromatography (Surface Measurement Systems Ltd., London, UK) with octane as solvent (*n* = 1). Samples were packed into silanized glass columns and stoppered using silanized glass wool at both ends. Dead volume was determined by methane injections. Retention times of probe molecules and methane were determined using a flame ionization detector (FID). Prior to measurement, samples were conditioned for 60 min at measurement settings of 30 °C, 0% relative humidity, and a 10 cm^3^/min nitrogen carrier gas flow. SSA was calculated from the isotherm of physical adsorption of octane molecules onto the solid’s surface in the pressure range (p/p0) from 0.05 to 0.35 according to Brunauer–Emmett–Teller [27,28].

#### 2.2.7. Scanning Electron Microscopy

Particle and surface morphology of the powders and respective granules were examined by scanning electron microscopy (SEM) via a LEO Gemini 1530 (Carl Zeiss AG, Oberkochen, Germany) with an acceleration voltage of 5.0 kV after sputtering the samples with a 10 nm platin coating to improve the electron conductivity. Samples were scanned with a magnification of 25× up to 20.000×.

#### 2.2.8. Tablet Compression

Compression of the powders and granules was performed with the Styl’One Evolution compaction simulator (Medelpharm, Beynost, France) using biplanar punches, with a diameter of 11.28 mm and a default compression profile without precompression at 20% compression speed. A force study was conducted with four maximum compression stresses varying from 50 to 200 MPa. 25 tablets for each powder blend and respective granules were produced at each compression pressure. Sample mass of 400 mg per tablet was weighed by Quantos dosing system QB1 (Mettler Toledo, Columbus, OH, USA) with a deviation <1% and powders were filled into the die manually. Punches and die were externally lubricated prior to every tablet production with magnesium stearate powder, which was applied with a brush.

#### 2.2.9. Tablet Characterization

Weight, thickness, diameter, and breaking force of the tablets (*n* = 10) were determined using a MultiCheck VI (Erweka GmbH, Langen, Germany) with a constant tablet breaking speed of 2.3 mm/s.

##### Tablet Tensile Strength

Diametral tablet tensile strength (TTS) was calculated according to Fell and Newton [29]:(2)$$\mathrm{Tablet}\;\mathrm{tensile}\;\mathrm{strength}\;\left(\mathrm{MPa}\right)=\frac{2\times F}{\pi \times d\times t}$$
with F as the tablet breaking force (N), d as tablet diameter (mm), and t as tablet thickness (mm).

##### Tablet Solid Fraction

The tablet solid fraction (SF) is directly correlated to tablet porosity (ε) as Solid fraction = 1 − ε and was calculated using the following equation with the tablet mass (m) and pycnometric density of the samples (ρ_p_).
(3)$$\mathrm{Solid}\;\mathrm{fraction}\;=\;\frac{4\times m}{\pi \times d^{2}\times t\times \rho_{p}}$$

##### Elastic Recovery

Based on compression force and punch displacement data, the in-die compression analysis was performed (*n* = 10). Elastic machine deformation was considered. Total work of compaction (TWC) was calculated as the integral of the force (F) over the distance (D) covered between the compact height at start of the force application (D_(0)_) and at maximum force (D_(Fmax)_) according to Çelik and Marshall [30].
(4)$$\mathrm{TWC}\;=\;\overset{D_{\left(\mathrm{Fmax}\right)}}{\underset{D_{\left(0\right)}}{\int }}F\times dD$$

Elastic recovery work (ERW) was determined by the integral of the force (F) over the distance (D) covered between compact height at maximum force (D_(Fmax)_) and compact height reached at the end of the compression force (D_(end)_).
(5)$$\mathrm{ERW}\;=\;\overset{D_{\left(\mathrm{end}\right)}}{\underset{D_{\left(\mathrm{Fmax}\right)}}{\int }}F\times dD$$

It should be noted that only upper punch data were included for TWC and ERW calculations, as the differences between upper and lower punch data were shown to be negligible in the compression profile used. With these parameters, the percentage ratio (ERW%) of ERW to TWC can be calculated as
(6)$$\mathrm{ERW}\%\;=\;\frac{\mathrm{ERW}}{\mathrm{TWC}}\times 100$$

##### Heckel Yield Pressure

Heckel analysis is an empirically derived method to describe a material volume reduction and plastic deformation under pressure. The Heckel equation is given as
(7)$$\ln\;(\frac{1}{1-\rho_{\mathrm{tab}}})\;=\;K\times P+A$$
with ρ_tab_ as the tablet’s relative density and P as applied compression pressure. By plotting ln (1/1 − ρ_tab_) versus the compression pressure P, the regression coefficients A and K can be determined from the intercept and the slope of the linear part of the curve. The yield pressure Py is the reciprocal of K and is commonly used to describe a powder’s viscoelastic behavior. Additionally, it is an indicator for the granule hardening phenomenon, one of the explanations for the loss in tabletability after roller compaction [31,32,33]. For this study, the Heckel yield pressure was determined with in-die tableting data (*n* = 10).

## 3. Results and Discussion

### 3.1. Impact of Polymorphism on Mechanical Material Characteristics

#### 3.1.1. Powder and Granule Characterization

Particle size distribution, bulk and tapped density, flowability as ffc, and specific surface area for β and δ mannitol powder and roller compacted granules are displayed in Table 1. The polymorphic form of the mannitol powders, granules, and respective tablets was monitored after all processing steps, and no polymorphic transformation occurred due to mechanical stress (Appendix A). Granulation is often performed for particle size enlargement, which was achieved for the β mannitol as well as the δ mannitol polymorph. An overall larger bulk and tapped density was observed in the granules compared to the respective powders as an additional typical effect and purpose of granulation. The β mannitol powder exhibits higher density than the δ mannitol powder and better flow properties than δ mannitol powder. After roller compaction, however, the density of the two polymorphs converged, and at the same time, the better flowability of β mannitol compared to δ mannitol was no longer evident. The granules exhibited decreased flowability compared to their powdery counterparts, despite their larger particle size distribution. In contrast to the results of Wagner et al., free flow for the granules was not achieved despite relatively high roller compaction forces [20]. From the increase in particle size after roller compaction, one could have expected smaller SSA of the granular particles compared to the powder. Instead, a larger surface area in the granules was observed, which could be attributed to the fact that mannitol is a brittle material [34,35] and extensive fracturing occurred during RC [36,37], resulting in higher surface roughness of the particles and increased surface area. This hypothesis could also be supported by the SEM images of the polymorphs (Figure 1), which showed an increased surface roughness of the granular particles. Therefore, a partial reason for the reduced flowability of the granules could be their higher surface area compared to the respective powders, leading to higher adhesive and cohesive forces. However, from the data in Table 1, it was also concluded that cohesive forces are not dependent on SSA alone, as δ mannitol had a higher SSA than β mannitol in powders as well as the granules, but δ mannitol only flows worse than β mannitol in the powders and not in the granules.

#### 3.1.2. Mechanical Characterization

The mannitol powders and granules showed pronounced sticking on the tableting punches and die during tablet production. Therefore, all tablets were externally lubricated with magnesium stearate. δ mannitol showed overall superior tableting properties compared to β mannitol, both in direct compression and after roller compaction. A better tabletability, meaning a higher tablet tensile strength (TTS) at the same compression pressure, was seen for δ mannitol powder and granules compared to β mannitol (Figure 2a). The larger specific surface area of δ mannitol seen in the powder and granules characterization constitutes a partial explanation for the better tabletability, as a larger SSA leads to a larger bonding area for the particles during tableting, consequently resulting in harder tablets [38,39]. This finding is in alignment with the research of Wagner et al. and Vanhoorne et al. [19,20]. The δ mannitol powder demonstrated a higher TTS compared to the respective granules despite the granules’ larger SSA, which could be an indicator for “work hardening” or “granule hardening” in δ mannitol granules, leading to a loss in tabletability [40,41]. This illustrates that finding the main influencing factors of a material’s mechanical properties and comparing manufacturing processes against each other is of high importance for formulation development, manufacturability, and troubleshooting, but also that one parameter alone is not sufficient to explain material behavior comprehensively.

Furthermore, the compactability plots (Figure 2b) demonstrate that at similar solid fraction, δ mannitol tablets reached higher TTS than β mannitol in direct compression and after RC. At a similar solid fraction, the δ mannitol powder showed slightly higher TTS to its respective granules and β mannitol exhibited comparable TTS of powder and granules, except for the tablets compressed at 150 MPa with a solid fraction 0.85. This clearly showed that the TTS differences between δ mannitol and β mannitol, and between δ mannitol powder and granules seen in the tabletability plot, are not due to differences in tablet solid fraction.

Yoshinari et al. and Vanhoorne et al. hypothesized that differences in the elastic–plastic behavior of β and δ mannitol are among the reasons for improved processability of δ mannitol in wet granulation and twin screw granulation [17,19]. Such differences were found in this study for roller compaction and direct compression, as the percentage of elastic recovery was greater in β mannitol than in δ mannitol in both process routes (Figure 3). A possible reason for this observation is, again, the specific surface area, as the larger SSA and the corresponding larger bonding area of δ mannitol result in stronger bonds within the tablet and thus less elastic recovery. It should be noted that for β mannitol the percentage of elastic recovery increased with compression pressure, whereas for δ mannitol it stayed constant over the complete pressure range, again indicating stronger bonds within the δ mannitol tablets. In both polymorphs, the granules exhibited higher elastic recovery compared to the respective powders despite their larger SSA. One reason could be a complex interplay between the granules’ higher specific surface area and the granule hardening effect, which is one explanation for the common loss in tabletability due to roller compaction [42]. A higher specific surface area of the granules compared to the respective powder would lead to a higher TTS due to an increased bonding area, but granule hardening, on the other hand, would reduce the TTS in the granules compared to the powder. These two impacts could counterbalance each other in β mannitol, whereas in δ mannitol the granule hardening could have a stronger impact on the granules than the increased specific surface area. SSA of brittle materials such as mannitol increases with compression pressure during the fracture phase, and new surfaces are exposed when the particles break [43]. One explanation could therefore be that β mannitol granules fractured more with increasing compression pressure than δ mannitol granules, leading to larger increase in SSA during compression and better compensation of the granule hardening effect. However, further research would be needed to verify this hypothesis.

An indicator for granule hardening is the yield pressure derived from Heckel analysis of the tableting data [31,32,33], as it is a parameter describing the material‘s ability to plastically deform. When granule hardening occurs, the material is less prone to plastically deform, and therefore an increase in yield pressure would be expected. An increase of yield pressure from powder to granules can therefore hint at granule hardening after roller compaction in a material. The yield pressure for both mannitols was higher in the granules compared to the respective powder, indicating granule hardening for both mannitols after roller compaction (Figure 4). δ mannitol powder displays a slightly lower yield pressure than β mannitol powder, whereas in the granules, both mannitols display similar yield pressures. As such, the increase in yield pressure is slightly higher for δ mannitol, suggesting that it could be more affected by the granule hardening effect. These findings support the earlier stated hypothesis—that granule hardening could counterbalance the positive effect of the granules’ higher specific surface area on TTS and that granule hardening could impact δ mannitol more strongly, leading to lower tabletability of the δ mannitol granules compared to the powder.

Another indication for the impact of the granule hardening phenomenon could possibly be seen in the loss in powder compressibility after roller compaction (Table 2). In the Manufacturing Classification System by Leane et al. [22], the loss in compressibility and limited compressibility is listed as critical property for roller compaction processes, and a correlation between the loss in compressibility after roller compaction and loss in tabletability and granule hardening has been described by Freitag et al. [44] and Santl et al. [45]. Therefore, despite strongly differing compression forces applied by the FT4 powder rheometer and in the tableting process, the FT4 powder compressibility could give an indication about the extent of granule hardening and possible loss in tabletability after roller compaction. For δ mannitol, a loss in powder compressibility of more than 8 % from powder to granules at 15 kPa compression force was recorded, whereas in β mannitol, the difference between powder and granules is minimal, with less than 2%. The reduced powder compressibility in δ mannitol could be a result of higher resistance against deformation due to granule hardening. The loss in compressibility of δ mannitol as opposed to β mannitol again supports the hypothesis that granule hardening impacts δ mannitol more strongly.

### 3.2. Impact of Particle Size on Mechanical Material Characteristics

#### 3.2.1. Powder and Granule Characterization

Particle size is a fundamental material characteristic that can influence many other powder properties. As such, the impact of particle size on the differences seen between β and δ mannitol was evaluated as part of the systematic assessment, presented by sieving β mannitol through a 180 µm sieve (180 β mannitol) and comparing it to the initial particle size distribution. Sieving adjusted the particle size of β mannitol powder to better match that of δ mannitol. However, after RC, similar granule sizes were observed despite the different initial particle sizes of the β mannitol powders. This indicates that roller compaction can compensate particle size differences in the powder and that granule size could be more profoundly impacted by the process settings and other material properties, which were not modified by the additional sieving step, than the initial particle size. The 180 β material showed a slightly smaller bulk and tapped density than the initial material and was still larger than the density of δ mannitol (Table 1). The powder compressibility of the 180 β mannitol powder is higher than for β mannitol powder, as there are less larger particles that could lead to tilting. Therefore, less voids could be created and a denser packing of the 180 β mannitol powder was reached (Table 2). In contrast to that, comparable powder compressibility was shown for both β mannitol granules, as they also displayed similar particle size after roller compaction. Therefore, it can be assumed that powder compressibility is dependent on particle size. Flowability of the 180 µm sieved β mannitol powder (FFC 6.56) and granules (FFC 5.43) was very similar to δ mannitol powder (FFC 6.21) and granules (FFC 5.31), which illustrates that the better flowability of the initial β mannitol powder was mainly due to its larger particle size (Table 1). Thus, as the differences of the polymorph powders in flowability could be mostly attributed to the different particle sizes of δ and β, it can be concluded that flowability can be excluded as a bias for differences in the mechanical properties of the two polymorphs after the sieving of β. By using two different particle sizes of the same β mannitol (initial material and 180 µm sieved), larger differences in the SSA values could have been expected than 0.23 m²/g vs. 0.30 m²/g in the powders and 0.94 m²/g vs. 0.81 m²/g in the granules, but both β mannitol demonstrate lower SSA than δ mannitol in powder and granules (Table 1). In powders, the initial material had lower SSA than the sieved material as opposed to the granules, where the initial material has a slightly higher SSA than the sieved material. This could be due to the fact that the initial material breaks more during roller compaction because of its larger particles, resulting in more accessible surfaces or surface roughness. It can therefore be stated that the particle size has an impact on many of the powder properties, albeit to different extents.

#### 3.2.2. Mechanical Characterization

The compactability plot (Figure 5) reveals almost identical TTS and SF for the tablets of the initial and 180 β mannitol. This is not surprising for the granules of both β mannitol samples, as a similar particle size after roller compaction was observed. For the powders, on the other hand, nearly the same TTS and SF values were determined despite their varying particle size distribution. Additionally, the powder sieving step did not have an effect on the elastic recovery percentage of β mannitol powder and granules (Figure 6). Initial and 180 β mannitol demonstrated higher elastic recovery than δ mannitol in the powders and granules. Yoshinari et al. presented the hypothesis that δ mannitol has improved tableting properties compared to β mannitol due to smaller particles that can be compacted more firmly [17]. In this study, the authors compared two particle sizes of the same β mannitol, which allows the systematic evaluation of the impact of particle size on the mechanical processability of mannitol. Similar mechanical properties during tableting were observed despite different particle sizes, which, in contrast to Yoshinari et al. [17], illustrates that other factors than particle size must be predominant in terms of the mechanical properties of mannitol. Even though the particle size impacted the powder and granule characteristics like flowability and density, the effect did not transfer to the mechanical properties of mannitol. Thus, the conclusion can be drawn that particle size plays only a marginal role for the mechanical properties compared to its different polymorphic forms, and consequently, other parameters must be crucial for the mechanical tableting properties such as surface properties. The similar mechanical properties of the two β mannitols, despite their differences in particle size, could be at least partially attributed to the comparable SSA, as this could lead to a comparable bonding area and therefore to similar tabletability and elastic recovery of the initial and 180 β mannitol. However, the difference in tablet tensile strength between δ mannitol compared to the two β mannitols is stronger than the difference in SSA, which again shows that SSA alone cannot describe particle bonding behavior and cohesive forces entirely, as described in Section 3.1.1.

### 3.3. Impact of Morphology and Preprocessing of Mannitol on Mechanical Material Characteristics

#### 3.3.1. Powder and Granule Characterization

By systematically characterizing the powder and granule properties of the two mannitol polymorphs, the impact of surface properties such as SSA on tablet manufacturability was highlighted. Hence, it was investigated whether the differences seen between the polymorphs can be mainly attributed to their crystal form or to particle properties such as surface and particle morphology. Therefore, the initial β and δ mannitol were compared to a spray granulated β mannitol, namely Parteck^®^ M200. 

The powder X-ray diffraction pattern confirmed that the spray granulated Parteck^®^ M200 consists of the β polymorph (Appendix A). The spray granulated β mannitol demonstrated a similar particle size to the initial β mannitol (Table 1). After RC, the differences were more pronounced, as Parteck^®^ M200 granules consisted of larger particles than the initial β mannitol, especially in the d50 and d90 values. Nevertheless, all β mannitol granules exhibited a smaller particle size than δ mannitol. Parteck^®^ M200 displayed the lowest tapped density compared to the unprocessed mannitols and also the least change in density before and after tapping (Table 1), implying that the spray granulated mannitol was already close to its optimal packing density before tapping and was not strongly compressed by tapping, thus suggesting a good flowability. The spray granulation of mannitol in Parteck^®^ M200 led to very good flow properties, as shown by its ffc value of over 37, which classifies it as free flowing. After RC, the flowability of Parteck^®^ M200 was reduced and was only slightly better compared to the other mannitols. All roller compacted granules were classified as easy flowing, with ffc values between 4 to 10 (Table 1). Due to the spray granulation manufacturing process, a large SSA of 2.80 m²/g for the Parteck^®^ M200 particles was generated. The roller compacted granules produced from Parteck^®^ M200 demonstrated the largest SSA of all samples (4.48 m²/g). This large surface area can be attributed to a high surface unevenness and roughness of the spherical particles, which was identified in the SEM images (Figure 1). The images also displayed that roller compaction enhanced the surface roughness even more, and it appeared that the needle-like structures on the powder particles fractured during granulation, leading to an increased surface area. The powder compressibility of the spray granulated Parteck^®^ M200 is under 5% over the complete pressure range, which is very low compared to the other mannitol powders (Table 2), again showing that after spray granulation an almost optimal packing density was reached for the bulk material. In contrast to the initial β and δ mannitol, an increase in powder compressibility after roller compaction was noticed for the spray granulated material, which can be attributed to the preprocessing step. After spray granulation the particles have spherical shape and an almost optimal packing density, leading to very low compressibility values, but during roller compaction fracturing of the particles occurs, and increasing surface roughness was seen. This could lead to tilting of the particles and higher cohesive forces that allowed for more cavities in the bulk, which then collapsed when compression pressure was applied. Parteck^®^ M200′s differing trend from raw spray granulated material to roller compacted granules in the FT4 powder compressibility compared to the other mannitols, and its immense loss in flowability after roller compaction indicated that preprocessing in the form of spray granulation not only strongly influences the raw material properties but can also considerably affect the impact of roller compaction and possibly other processes on the material properties.

#### 3.3.2. Mechanical Characterization

Parteck^®^ M200 raw material and roller compacted granules displayed the highest TTS in the tabletability plots (Figure 7a). At a similar solid fraction, tablets produced by spray granulated Parteck^®^ M200 presented the highest TTS by far compared to initial β and δ mannitol (Figure 7b). This is likely a result of the higher SSA of the spray granulated material leading to a higher bonding area and therefore generating stronger tablets. Therefore, it is indicated that the disadvantages of β polymorph compared to δ mannitol seen during RC and tableting can be overcome by spray granulation of β mannitol and modifying particle surface and morphology. In contrast to the initial β, the Parteck^®^ M200 roller compacted granules demonstrate a strong decrease in TTS compared to the respective raw material and exhibited a very pronounced granule hardening, which was indicated by the increase in the Heckel yield pressure. It is noteworthy that the increase of yield pressure from Parteck^®^ M200 raw material to granules is significantly higher than seen in initial β and δ mannitol (Figure 8). This implies a stronger increase in resistance against plastic deformation after roller compaction in Parteck^®^ M200, and thus a stronger granule hardening effect compared to initial β and δ mannitol can be deduced. One possible explanation for the pronounced loss in tabletability after roller compaction of Parteck^®^ M200 could be that the distinct porous structure of the particles is densified during roller compaction, leading to less deformable material, which would consequently lead to a higher porosity and less bonding area in the tablets, as seen at high compression pressures in the compactability plot (Figure 7b) and therefore to a lower tabletability than the raw spray granulated material [46,47]. However, further experiments are needed for verification. The lower TTS of roller compacted Parteck^®^ M200 granules compared to the raw material, despite having a higher surface area and thus higher bonding area, also supports the hypothesis already established in 0—that an increased surface area of the granules leading to a higher TTS and lower elastic recovery was opposed by granule hardening, decreasing the TTS. This was also hypothesized by Wagner et al. [20], and could now be substantiated with data in this study. By using the Heckel yield pressure as a parameter to characterize the granule hardening effect and considering the elastic–plastic properties of powder and respective granules of the polymorphs, the hypothesis was further elaborated and strengthened.

The elastic recovery percentage of Parteck^®^ M200 was lower than for the other β mannitol samples and comparable to δ mannitol powder. The reduced elastic recovery percentage and the high surface area, and thus high bonding area, could explain the high TTS of Parteck^®^ M200 (Figure 9). Although the Parteck^®^ M200 raw material displays comparably low elastic recovery, the roller compacted Parteck^®^ M200 revealed the overall highest elastic recovery percentage, which supports the observation that roller compaction had the strongest impact on the spray granulated material. The high elastic recovery of the Parteck^®^ M200 granules provides a partial rationale for the lower TTS compared to the tablets from the raw material, as a higher elastic recovery indicates less strong interparticle bonds within the tablet. Nevertheless, the high specific surface area and the resulting high bonding area of the Parteck^®^ M200 roller compacted granules still enable more interparticle bonds and a higher TTS than in the initial β and δ mannitol.

With this comparison of spray granulated and initial mannitol, it was shown that the preprocessing of the material via spray granulation prevails over the advantages of δ mannitol seen during tableting. Especially in direct compression, highest tabletability and low elastic recovery were detected for Parteck^®^ M200. This superiority is attributable to the changed morphology and strongly increased surface area due to spray granulation, leading to an increased bonding area.

### 3.4. Direct Compression vs. Roller Compaction

In direct compression, the tablet is produced directly from the powder, whereas in roller compaction, the powder is granulated before tableting so that granules are manufactured as an intermediate product. Therefore, the process routes direct compression and roller compaction were contrasted in this study through a systematic comparison of powder, respective granules, and the tablets produced from each. 

Clear differences in the absolute characteristic values were found for almost all evaluated attributes between roller compaction and direct compaction. As roller compaction is often performed for particle size enlargement, it is not surprising that the granules display larger particle size than the respective powders (Table 1). In the powders, the particle size of δ mannitol was smaller compared to β mannitol, whereas in the granules this observation was reversed and δ mannitol exhibited the larger granules. The phenomenon, that smaller primary particles lead to larger roller compacted granules was already described by Herting and Kleinebudde [48]. Roller compaction resulted in different extents of loss in tabletability, increase in yield pressure, and an increase in elastic recovery percentage compared to direct compression of the mannitols, as discussed under Section 3.1.2 and Section 3.3.2. Roller compaction impacted the spray granulated β material the most, δ mannitol was less pronounced, and the initial β mannitol was impacted the least. The distinct effect of roller compaction on the tabletability and yield pressure of the spray granulated material indicates that the extent of loss in tabletability is strongly dependent on particle characteristics such as morphology, surface characteristics, and porosity. This hypothesis is also supported by the research of Grote et al., where the impact of morphology of dibasic calcium phosphate on tabletability after roller compaction was demonstrated [49].

It should be particularly emphasized that even though the absolute measurement values differ between the roller compaction and direct compression process route, the overall trends observed often still remain the same. This is not only the case for powder/granules analytics like specific surface area, which increased from β mannitol to δ mannitol to spray granulated β mannitol in powder and granules, or the tapped density, which decreased from β mannitol to δ mannitol to spray granulated β mannitol, as it was also seen in the mechanical tablet characteristics. The mannitols exhibited the same trend in tabletability and compactability in both processing routes—roller compaction and direct compression. Even though a loss in tabletability from powder to granules was visible in the spray granulated β mannitol, TTS follows the trend spray granulated β mannitol > δ mannitol > β mannitol in direct compression and tableting after roller compaction (Figure 7). Although the Heckel yield pressure increased from directly compressed to roller compacted material, spray granulated β mannitol had the highest yield pressure in both process routes (Figure 8). 

Obtaining the same trends in direct compression and roller compaction illustrates that the material specific characteristics still clearly influence the powder behavior after roller compaction. Even though roller compaction can affect the material properties, the comparison of these two processing routes showed that particle properties of the initial material were more critical for the bulk properties and processability of the mannitols than the process routes of roller compaction or direct compression itself. Therefore, particle properties in the early development stage should be optimized by particle engineering, followed by the selection of an appropriate manufacturing process.

## 4. Conclusions

The multidimensional effects of polymorphism, particle size, surface properties, and the process route on the mechanical behavior of mannitol for oral solid dosage forms were elucidated through systematic powder characterization. Better processability of δ mannitol compared to β mannitol in direct compression and roller compaction could mainly be attributed to its higher surface area and less to its polymorphic form. This was supported by the fact that superior TTS of β mannitol compared to δ mannitol was reached by preprocessing β mannitol by spray granulation, which resulted in a greatly increased surface area and, consequently, in a higher bonding area. Thus, surface area was identified as one of the reasons for good tabletability and was highlighted as a potential key material attribute for processability of the two polymorphs. However, the study also emphasized that TTS is not solely dependent on a single powder characteristic. The granules exhibited the same or lower TTS than the respective powders despite higher SSA, which could possibly be explained by compensation of the higher bonding area due to the granule hardening phenomenon after roller compaction.

Particle size did not demonstrate a significant impact on the tableting characteristics of the mannitol polymorphs, when two different particle sizes of the same β mannitol were compared. Other factors than the particle size must therefore prevail in terms of the mechanical properties of the polymorphs.

The direct comparison of two process routes for oral solid dosage forms, direct compression, and roller compaction highlighted that initial powder and particle characteristics had a stronger impact on tablet manufacturability of the mannitol polymorphs than the process route, as similar trends in the powder and respective granules were evident in tableting.

With this study, a profound insight into the mechanical characteristics of mannitol was gained, and the importance of systematic powder and particle characterization of excipients early in the development stage for oral solid dosage forms was emphasized.

## Figures and Tables

**Figure 1 pharmaceutics-14-02128-f001:**
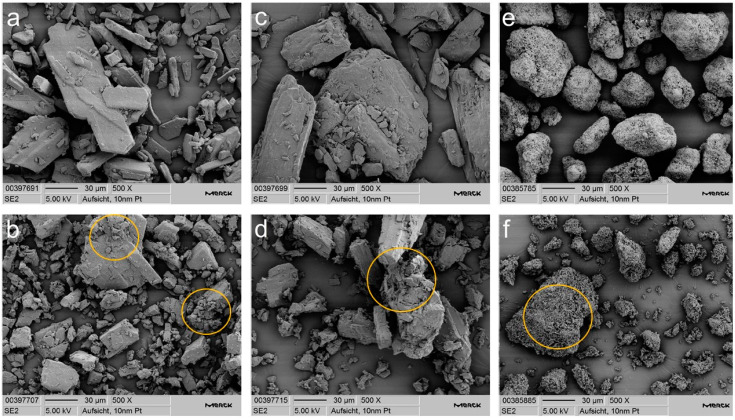
Scanning electron microscopy images of (**a**) δ mannitol powder, (**b**) δ mannitol granules, (**c**) β mannitol powder, (**d**) β mannitol granules, (**e**) spray granulated Parteck M200, and (**f**) spray granulated Parteck M200 after roller compaction at 500× magnification. Regions with high surface roughness after roller compaction marked in yellow.

**Figure 2 pharmaceutics-14-02128-f002:**
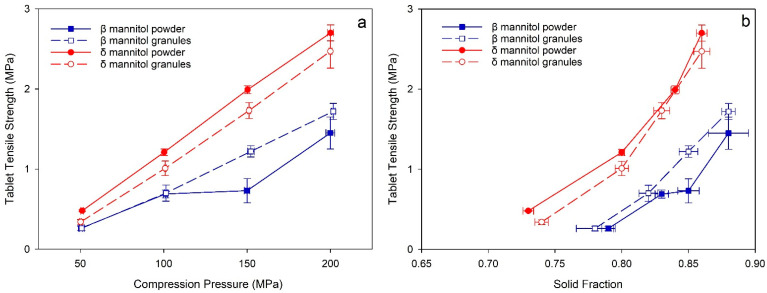
(**a**) Tabletability and (**b**) compactability plots of δ mannitol and β mannitol powder and granules. Arithmetic means of *n* = 10 ± S.D.

**Figure 3 pharmaceutics-14-02128-f003:**
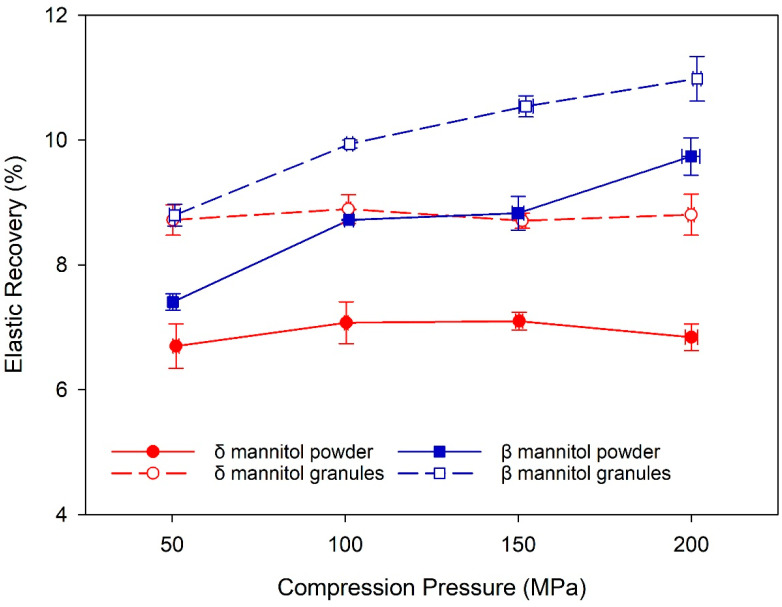
The percentage of elastic recovery for δ and β mannitol powder and their respective granules. Arithmetic means of *n* = 10 ± S.D.

**Figure 4 pharmaceutics-14-02128-f004:**
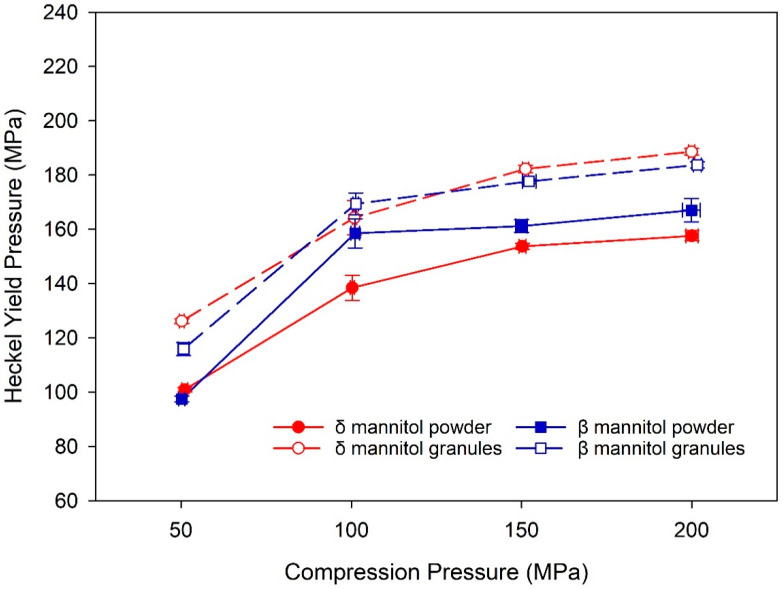
Heckel Yield pressure of δ and β mannitol powder and their respective granules. Arithmetic means of *n* = 10 ± S.D.

**Figure 5 pharmaceutics-14-02128-f005:**
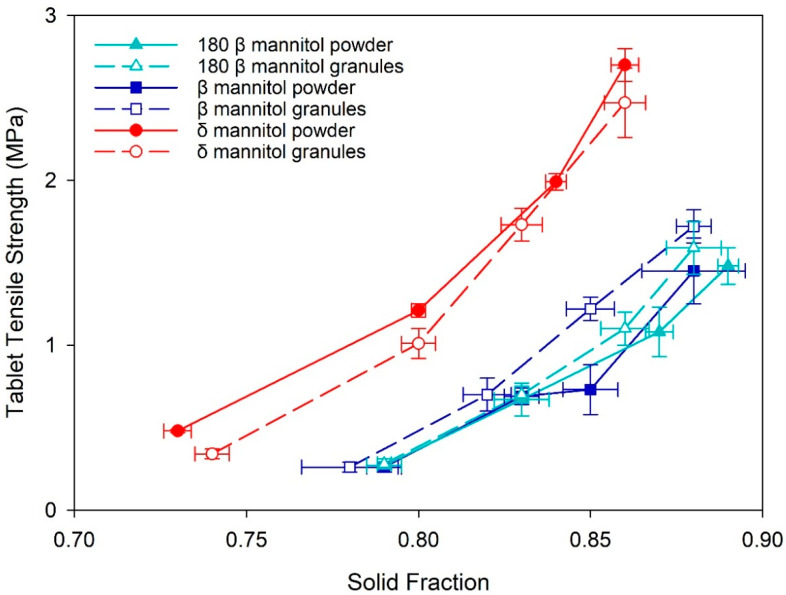
Compactability of δ mannitol, β mannitol, and 180 µm sieved β mannitol powder and granules. Arithmetic means of *n* = 10 ± S.D.

**Figure 6 pharmaceutics-14-02128-f006:**
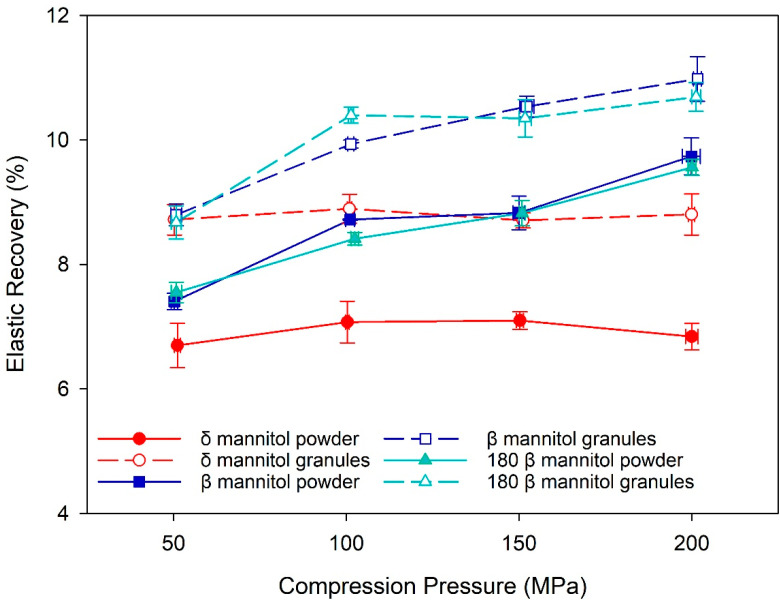
The percentage of elastic recovery for δ mannitol, β mannitol, and 180 µm sieved β mannitol powder and granules. Arithmetic means of *n* = 10 ± S.D.

**Figure 7 pharmaceutics-14-02128-f007:**
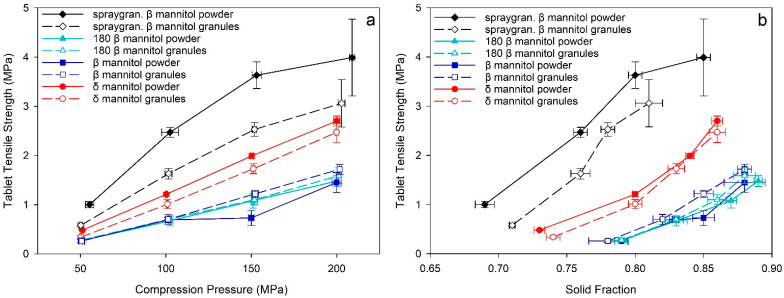
(**a**) Tabletability and (**b**) compactability plot of δ and β mannitol, β mannitol sieved through 180 µm sieve, and spray granulated β mannitol and their respective roller compacted granules. Arithmetic means of *n* = 10 ± S.D.

**Figure 8 pharmaceutics-14-02128-f008:**
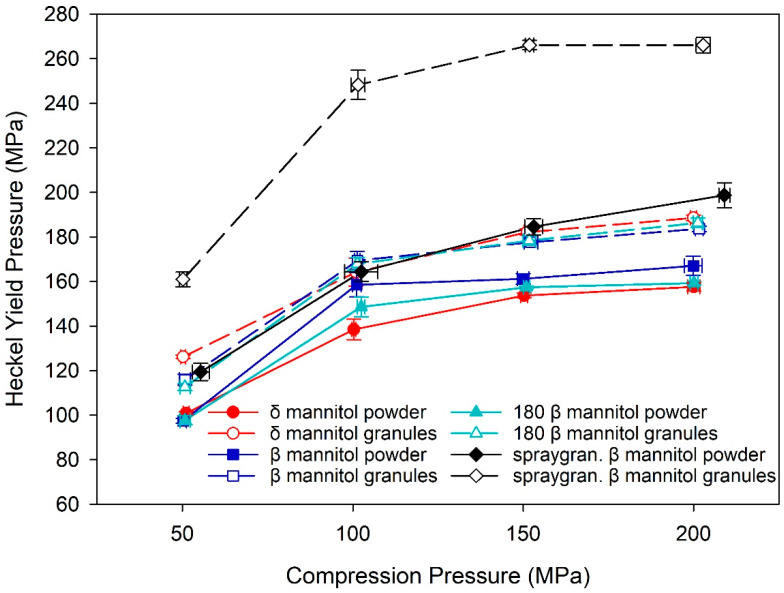
Heckel yield pressure of δ and β mannitol, β mannitol sieved through 180 µm sieve, and spray granulated β mannitol and their respective roller compacted granules. Arithmetic means of *n* = 10 ± S.D.

**Figure 9 pharmaceutics-14-02128-f009:**
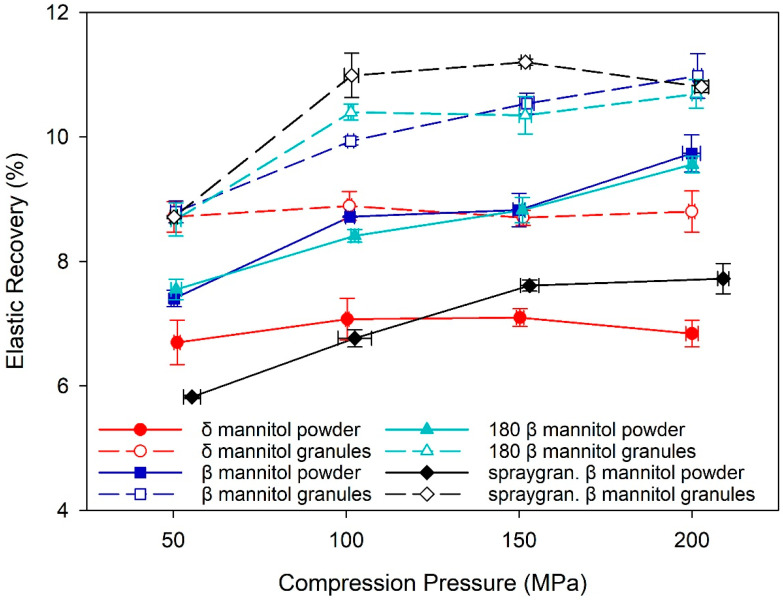
The percentage of elastic recovery for δ and β mannitol, β mannitol sieved through 180 µm sieve and spray granulated β mannitol and their respective roller compacted granules. Arithmetic means of *n* = 10 ± S.D.

**Table 1 pharmaceutics-14-02128-t001:** Particle size distribution, bulk and tapped density, flowability as ffc value and specific surface area of δ mannitol, β mannitol, 180 µm sieved β mannitol, and spray granulated β mannitol powder and the respective granules. Arithmetic means of *n* = 3 ± S.D. except for SSA (*n* = 1).

Polymorph		Particle Size Distribution (µm)			Density (g/mL)		ffc	SSA (m²/g)
		d10	d50	d90	ρ_b_	ρ_t_		
**Powder**	δ mannitol	22 ± 0	57 ± 1	171 ± 2	0.493 ± 0.007	0.658 ± 0.001	6.21 ± 0.19	0.44
	β mannitol	37 ± 3	127 ± 4	271 ± 3	0.598 ± 0.004	0.744 ± 0.004	8.40 ± 0.24	0.23
	180 β mannitol	36 ± 2	110 ± 3	205 ± 1	0.565 ± 0.001	0.726 ± 0.003	6.56 ± 0.16	0.30
	spray granulated β mannitol	73 ± 5	149 ± 1	233 ± 1	0.542 ± 0.003	0.608 ± 0.003	37.33 ± 8.16	2.80
**Granules**	δ mannitol	31 ± 3	671 ± 42	1087 ± 22	0.613 ± 0.005	0.776 ± 0.009	5.31 ± 0.22	1.80
	β mannitol	22 ± 0	164 ± 18	725 ± 26	0.649 ± 0.016	0.799 ± 0.006	4.94 ± 0.20	0.94
	180 β mannitol	32 ± 9	160 ± 34	671 ± 22	0.631 ± 004	0.791 ± 0.007	5.43 ± 0.58	0.81
	spray granulated β mannitol	107 ± 38	480 ± 41	879 ± 47	0.544 ± 0.009	0.647 ± 0.004	7.31 ± 0.16	4.48

**Table 2 pharmaceutics-14-02128-t002:** Powder compressibility of δ mannitol, β mannitol, 180 µm sieved β mannitol, spray granulated β mannitol powder, and the respective granules. Arithmetic means of *n* = 3 ± S.D.

Compression Pressure (kPa)	Compressibility (%)							
	Powder				Granules			
	δ man.	β man.	180 β man.	Spray Gran β man.	δ man.	β man.	180 β man.	Spray Gran β man.
**0.5**	7.3 ± 3.0	3.3 ± 0.5	4.3 ± 0.4	1.2 ± 0.1	4.2 ± 0.3	3.8 ± 0.1	5.2 ± 2.0	3.4 ± 0.3
**1**	8.5 ± 2.8	4.8 ± 0.4	5.9 ± 0.4	1.4 ± 0.1	5.5 ± 0.3	5.2 ± 0.1	6.4 ± 2.0	4.7 ± 0.2
**2**	13.5 ± 2.4	7.7 ± 0.4	9.5 ± 0.5	1.9 ± 0.1	8.2 ± 0.4	8.0 ± 0.2	9.2 ± 2.1	6.9 ± 0.1
**4**	18.0 ± 2.4	10.4 ± 0.4	12.9 ± 0.6	2.5 ± 0.2	11.0 ± 0.7	11.1 ± 0.2	12.3 ± 2.2	9.4 ± 0.2
**6**	20.3 ± 2.5	11.7 ± 0.5	14.5 ± 0.6	2.8 ± 0.2	12.6 ± 0.8	12.7 ± 0.2	13.9 ± 2.2	10.8 ± 0.2
**8**	21.8 ± 2.5	12.7 ± 0.5	15.7 ± 0.6	3.2 ± 0.2	13.6 ± 0.9	13.9 ± 0.1	15.2 ± 2.2	11.7 ± 0.3
**10**	22.8 ± 2.5	13.5 ± 0.4	16.7 ± 0.6	3.4 ± 0.2	14.5 ± 1.0	14.9 ± 0.1	16.1 ± 2.3	12.5 ± 0.3
**12**	23.7 ± 2.5	14.2 ± 0.4	17.5 ± 0.5	3.7 ± 0.3	15.2 ± 1.0	15.7 ± 0.2	16.9 ± 2.3	13.2 ± 0.3
**15**	24.8 ± 2.5	15.0 ± 0.5	18.4 ± 0.5	4.0 ± 0.3	16.2 ± 1.1	16.7 ± 0.1	18.0 ± 2.3	14.0 ± 0.4

## Data Availability

Data is contained within the article or Appendix A.

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
