# Peer review of "Understanding the Multidimensional Effects of Polymorphism, Particle Size and Processing for D-Mannitol Powders"

_pharmaceutics, 2022, doi:10.3390/pharmaceutics14102128_

Round 1

Reviewer 1 Report

The current study focuses on understanding the effect of particulate solid-state properties such as particle size, morphology, specific surface area etc. on the processing of different polymorphs of mannitol. The systematic approach adopted by the authors is good. I only have some minor suggestions listed below:

1.     Line 46: The meaning of the statement is not clear. Recommend modifying the sentence.

2.     Page 8 Table 1: It is unclear as to why the particle size of the sieved b mannitol powder and sieved b mannitol granules is significantly different. Wouldn’t sieving make the particle size in both cases more similar? Do the authors mean that the sample retained on the sieve is used for further analysis? I recommend explaining this in the material and method section for clarity. 

3.     Page 9 line 259: The authors claim that the SEM images show an increased roughness of the granular particles. However, it is not evident from the SEM images. Can authors provide more magnified images and maybe highlight the regions in the granules that show increased roughness to make it easier for the reader to follow?

4.     Page 13 line 347: The authors need not mention the title of the table in the text.

5.     Page 14 line 372: See comment number 3.

6.     Page 21 line 522: Throughout the manuscript, the authors have discussed the differences between the mannitol powders (obtained commercially) and granules (obtained after roller compaction). The data for direct compression is not presented in the text. It is not clear as to how authors are making comparison between direct compression and roller compaction as processing methods in Section 3.4.

Author Response

Dear reviewer,

Thank you for your time reading and reviewing our manuscript. We appreciate your comments and suggestions, which are discussed in the following section. The respective amendments were made to the manuscript.

The current study focuses on understanding the effect of particulate solid-state properties such as particle size, morphology, specific surface area etc. on the processing of different polymorphs of mannitol. The systematic approach adopted by the authors is good. I only have some minor suggestions listed below:

1. Line 46: The meaning of the statement is not clear. Recommend modifying the sentence. 

Thank you for bringing this point to our attention. As recommended, the sentence was rephrased to better elaborate the meaning of the statement.

2. Page 8 Table 1: It is unclear as to why the particle size of the sieved bmannitol powder and sieved b mannitol granules is significantly different. Wouldn’t sieving make the particle size in both cases more similar? Do the authors mean that the sample retained on the sieve is used for further analysis? I recommend explaining this in the material and method section for clarity. 

Thank you for this important remark. As suggested, the sieving step was further elaborated in the materials and methods section to highlight that only the initial β mannitol powder was sieved through the 180 µm sieve to compare the impact of different initial powder particle sizes on granules and tablet properties. The cumulative particle size curves were also added as Supplementary Material S1 to deeper illustrate the impact of the sieving step of the β mannitol powder.

3. Page 9 line 259: The authors claim that the SEM images show an increased roughness of the granular particles. However, it is not evident from the SEM images. Can authors provide more magnified images and maybe highlight the regions in the granules that show increased roughness to make it easier for the reader to follow?

and

5. Page 14 line 372: See comment number 3.

Thank you for raising this important point. You are right, images using larger magnification would better demonstrate the increased surface roughness in the granules. Unfortunately when using a larger magnification the differences in particle shape between the different polymorphs, especially the spherical Parteck M200 particles compared to the initial β and δ material, would no longer be so recognizable. Therefore, as recommended, the regions in the granules where the surface roughness is especially well displayed were highlighted.

4. Page 13 line 347: The authors need not mention the title of the table in the text.

Thank you for your comment, the text was adjusted as recommended.

6. Page 21 line 522: Throughout the manuscript, the authors have discussed the differences between the mannitol powders (obtained commercially) and granules (obtained after roller compaction). The data for direct compression is not presented in the text. It is not clear as to how authors are making comparison between direct compression and roller compaction as processing methods in Section 3.4.

Thank you for this important observation. As recommended, section 3.4. has been adapted to better elaborate the comparison between the processes direct compaction and roller compaction.

Kind regards,

The author

Reviewer 2 Report

In this work, the multidimensional effects of polymorphism, particle size, surface properties as well as the process route on the mechanical behavior of mannitol for oral solid dosage forms were elucidated through systematic powder characterization. Better processability of δ mannitol compared to β mannitol in direct compression and roller compaction could mainly be attributed to its higher surface area and less to its polymorphic form. This was supported by the fact, that superior TTS of β mannitol compared to δ mannitol was reached by preprocessing β mannitol by spray granulation, which resulted in a greatly increased surface area and consequently to a higher bonding area. Moreover, the granules exhibited the same or lower TTS than the respective powders despite higher SSA, which could possibly be explained by compensation of the higher bonding area due to the granule hardening phenomenon after roller compaction. Overall, with this study, a profound insight into the mechanical characteristics of mannitol was gained and it emphasized the importance of systematic powder and particle characterization of excipients early in the development stage for oral solid dosage forms. This study is quite well organized and worth to publish in Pharmaceutics. Before accepted, please check the following general concerns.

1. Three polymorphs (α, β, δ) and a monohydrate of mannitol are known, of which β is the thermodynamically most stable form and as such most frequently used. People may understand the δ polymorph  is thermodynamically metastable, but authors should also give the general discussion about β and δ polymorph (Introduction).

2. For power compressibility tests, are there any standard methods? I search the relevant references, seems less study tend to use FT4 powder rheometer (Freeman Technology, UK). Authors are recommended to cite some references in the experiment section here.

3. Format of the references. Ref. 2/11/32/37/41/46: no DOI links…

Author Response

Dear reviewer,

Thank you for your time reading and reviewing our manuscript. We appreciate your comments and suggestions, which are discussed in the following section. The respectective amendments were made to the manuscript.

In this work, the multidimensional effects of polymorphism, particle size, surface properties as well as the process route on the mechanical behavior of mannitol for oral solid dosage forms were elucidated through systematic powder characterization. Better processability of δ mannitol compared to β mannitol in direct compression and roller compaction could mainly be attributed to its higher surface area and less to its polymorphic form. This was supported by the fact, that superior TTS of β mannitol compared to δ mannitol was reached by preprocessing β mannitol by spray granulation, which resulted in a greatly increased surface area and consequently to a higher bonding area. Moreover, the granules exhibited the same or lower TTS than the respective powders despite higher SSA, which could possibly be explained by compensation of the higher bonding area due to the granule hardening phenomenon after roller compaction. Overall, with this study, a profound insight into the mechanical characteristics of mannitol was gained and it emphasized the importance of systematic powder and particle characterization of excipients early in the development stage for oral solid dosage forms. This study is quite well organized and worth to publish in Pharmaceutics. Before accepted, please check the following general concerns.

1. Three polymorphs (α, β, δ) and a monohydrate of mannitol are known, of which β is the thermodynamically most stable form and as such most frequently used. People may understandthe δ polymorph  is thermodynamically metastable, but authors should also give the general discussion about β and δ polymorph (Introduction).

Thank you for pointing this out. As recommended, a more general comparison of the polymorphs was added to the introduction.

2. For power compressibility tests, are there any standard methods? I search the relevant references, seems less study tend to use FT4 powder rheometer (Freeman Technology, UK). Authors are recommended to cite some references in the experiment section here.

Thank you for this important remark! As recommended, the established protocols for the powder compressibility on which our method is based were referenced.

3. Format of the references. Ref. 2/11/32/37/41/46: no DOI links…

Thank you for this observation. As recommended, the missing DOI links were added to the references.

Kind regards,

The author